# Spatio-Temporal Variability of Methane Fluxes in Boreo-Nemoral Alder Swamp (European Russia)

**Tamara V. Glukhova** [1,*,†]**, Danil V. Ilyasov** [2,*,†]**, Stanislav E. Vompersky** [1]**, Gennady G. Suvorov** [1]**, Alla V. Golovchenko** [3]**, Natalia A. Manucharova** [3] **and Alexey L. Stepanov** [3]

[1] Laboratory of Wetland Studies, Institute of Forest Science, Russian Academy of Sciences, Sovetskaya 21, 143030 Uspenskoe, Moscow Oblast, Russia; root@ilan.ras.ru (S.E.V.); suvorov@ilan.ras.ru (G.G.S.)

[2] Laboratory of Ecosystem-Atmosphere Interactions of the Mire-Forest Landscapes, Yugra State University, Chekhova 16, 628012 Khanty-Mansiysk, Russia

[3] Department of Soil Science, Lomonosov Moscow State University, 1–12, GSP-1, Leninskie Gory, 119991 Moscow, Russia; golovchenko.alla@gmail.com (A.V.G.); manucharova@mail.ru (N.A.M.); stepanov_aleksey@mail.ru (A.L.S.)

[*] Correspondence: glutam@mail.ru (T.V.G.); d_ilyasov@ugrasu.ru (D.V.I.)

[†] These authors contributed equally to this work.

**Abstract:** In 1995–1998 and 2013–2016, we measured methane fluxes (1Q-median-3Q, mgC m$^{-2}$ h$^{-1}$) in the Petushikha black alder swamp of the boreo-nemoral zone of European Russia. At microelevations (EL sites), flat surfaces (FL), microdepressions (DEP), and water surfaces of streams and channels (STR) sites, the fluxes comprised 0.01–0.03–0.09, 0.02–0.06–0.19, 0.04–0.14–0.43, and 0.10–0.21–0.44, respectively. The biggest uncertainty of methane fluxes was caused by seasonal variability (the level of relative variability of fluxes is a nonparametric analogue of the coefficient of variation) which comprised 144%, then by spatial variability—105%, and the smallest by interannual variability—75%. Both spatial and temporal variability of methane fluxes at different elements of the microrelief is heterogeneous: the most variable are communities that are "unstable" in terms of hydrological conditions, such as FL and DEP, and the least variable are the most drained EL and the most moistened STR ("stable" in terms of hydrological conditions). The obtained data on the fluxes and their spatial and temporal variability are consistent with the literature data and can be used to optimize the process of planning studies of the methane budget of "sporadic methane sources", such as waterlogged forests. This is especially relevant for an adequate assessment of the role of methane fluxes in the formation of the waterlogged forests carbon budget and a changing climate.

**Keywords:** waterlogged forests; methane emission; long-term monitoring; water table level





## 1. Introduction

Methane is the second most important greenhouse gas [1]: its contribution to the increase in the average atmospheric temperature is 13%–25%, considering the 100-year global warming potential [2–4]. The largest natural source of methane flux into the atmosphere, estimated at 110–170 Tg CH$_4$ yr$^{-1}$ [5,6], is wetlands. Only boreal wetlands located mainly between 50 and 70° N are responsible for the atmospheric influx of 25 to 65 Tg CH$_4$ yr$^{-1}$ [5,7–9].

The study of the mechanisms of functioning of wetlands as components of the atmospheric methane budget is necessary for a deeper understanding of their role in the process of climate change [10–12]. Modern estimates of the methane budget of wetlands are based on many direct (field) measurements and mathematical models [1,13,14]. However, different types of wetlands have not been studied with the same degree of detail, since attention has been primarily paid to the most widespread wetlands, i.e., those that undoubtedly contribute the most to the methane budget of wetlands. In the boreal and temperate zones, such wetlands are forested and unforested bogs and fens, and in the

tropical zone—swamps [6,7,10]. These ecosystems are characterized by constant abundant moistening from atmospheric precipitation or surface and ground waters, the relative permanence of the formation of anaerobic conditions, and, therefore, of the stable methane emissions [4,8].

However, for a full understanding of the role of wetlands in the global methane budget, it is also necessary to consider such ecosystems, which can be characterized by significant variability of $CH_4$ fluxes in space or time; that is, a periodic spatial or temporal shift of methane uptake to its emission. Examples of such ecosystems are periodically flooded floodplains, waterlogged forests of the boreal zone, forests with periodic excessive moistening, etc. (hereinafter referred to as sporadic sources—SSs).

Despite the impermanence of SSs as methane sources, $CH_4$ fluxes can vary there from negative values to a few tens of mgC-$CH_4$ m$^{-2}$ h$^{-1}$; for example, from 0.1 to 12.5 [15–17] in floodplains; from 0.2 to 8.3 [18–21] in waterlogged forests; and from 0.3 to 3.8 [22,23] even in automorphic forests characterized by only a periodic rise in groundwater level. These methane fluxes are comparable to, and under certain conditions (combination of soil moisture and soil temperature), even exceed those observed in bogs and fens [8,21,24–30].

Thus, SSs as sources of methane can potentially make a significant contribution to the regional and perhaps global methane budget. At the same time, the contribution of waterlogged and periodically waterlogged forests may be more significant due to their wide distribution. For example, in Russia the area of waterlogged forests is estimated at 24 to 111 million hectares [31,32]; waterlogged forests are also widespread in other countries of the boreal zone such as Finland, Sweden, Canada, USA, and—as was noted earlier—the tropical zone [33,34].

Swamps (constantly waterlogged forests) are characterized by extremely high heterogeneity of methane fluxes both in space and time, which makes it difficult to obtain adequate estimates of annual emissions. This is due to specific physicochemical (abundance of nutrients, acidity, redox conditions), hydrological (periodic flooding and drying), topological (microrelief and nanorelief) and other conditions, which significantly complicates their study and leads to ambiguous results, especially for short-term field studies. Therefore, obtaining adequate estimates of methane fluxes in waterlogged and/or periodically waterlogged forests is impossible without considering their seasonal, interannual, and spatial variability [18–23].

Swamp forests dominated by the black alder *Alnus glutinosa* (L.) Gaertn. in the tree layer are widespread in the European part of Russia, especially in the north of the Russian Plain. Black alder forests occupy 0.9 million hectares of this territory [35]. The northern border of their distribution runs through central Karelia (middle taiga) [36], and the southern border reaches the forest-steppe zone [37]. Black alder forests spread from the western border of Russia to the Ural Mountains [38] and further in the Trans-Urals to Tobol [35].

In addition to Russia, black alder forests grow predominantly in Belarus, occupying an area of 660 thousand hectares [39], in Ukraine on 130 thousand hectares [40], and to a certain extent in Lithuania, Latvia [41], and Estonia [42,43]. They are also found in Poland [44], North America [45,46], and Germany [47]. They usually occupy small (up to 200 hectares [37]) areas and are associated with such habitats as the foothills of slopes, near-terrace floodplains of rivers, riverbanks, lakes, and bog shores. They grow in places with abundant flow-through moistening and represent one of the stages of eutrophic swamping. Black alder forests are found not only in swamps, but also in dry habitats, with mesophilic species in the grass layer [48].

A large number of publications are devoted to the characteristics of black alder forest swamps, conditions of their formation, geographical distribution, geomorphological predisposition, structure and species diversity of plant communities [36,40,48–54]. However, there are few publications on the role of black alder forest swamps in the cycle of the main biogenic elements in the carbon cycle [43,47]. Black alder swamps are the least or completely unstudied biogeocenoses in this respect; this also applies to fluxes of greenhouse

gases (in particular, $CH_4$) which are an important indicator of the carbon balance of these ecosystems [55].

Unfortunately, in Russia studies of $CH_4$ fluxes from soils of waterlogged forests, including black alder ones, are extremely limited (measurements were performed without reference to seasonal, interannual, and spatial variability of conditions) and were carried out mainly in Western Siberia [30,56,57].

The aim of this work is the assessment of spatio-temporal variability of methane fluxes in a tall-herb and fern black alder forest located in the boreo-nemoral zone of European Russia [58].

## 2. Materials and Methods

### 2.1. Study Location

The studies were carried out during the summer and autumn periods of 1995–1998 and 2013–2016 on alder swamp "Petushikha" (56°10′15″ N, 32°08′16″ E) in the southern taiga zone of European Russia (Figure 1). The period from June to October in the southern taiga corresponds to summer (June–August) and the first half of autumn (September–October). The peatland "Petushikha" was under the influence of a powerful ecological factor throughout the history of its development—abundant water–mineral supply due to alluvial slope waters from the moraine hills adjacent to the swamp [59]. There is a swamp microlandscape closer to the periphery of the swamp, with variable flow-through moistening and abundant water–mineral supply. A virgin black alder forest with an area of 7.4 hectares has formed in a flat depression at the edges of moraine hills. The water–mineral supply is comprised of atmospheric, soil-ground, and alluvial slope (of transit nature) waters; groundwater wedging out from the above-mooric horizons is observed on the surface of the swamp. The alder swamp is drained by a stream that does not have a well-defined watercourse.

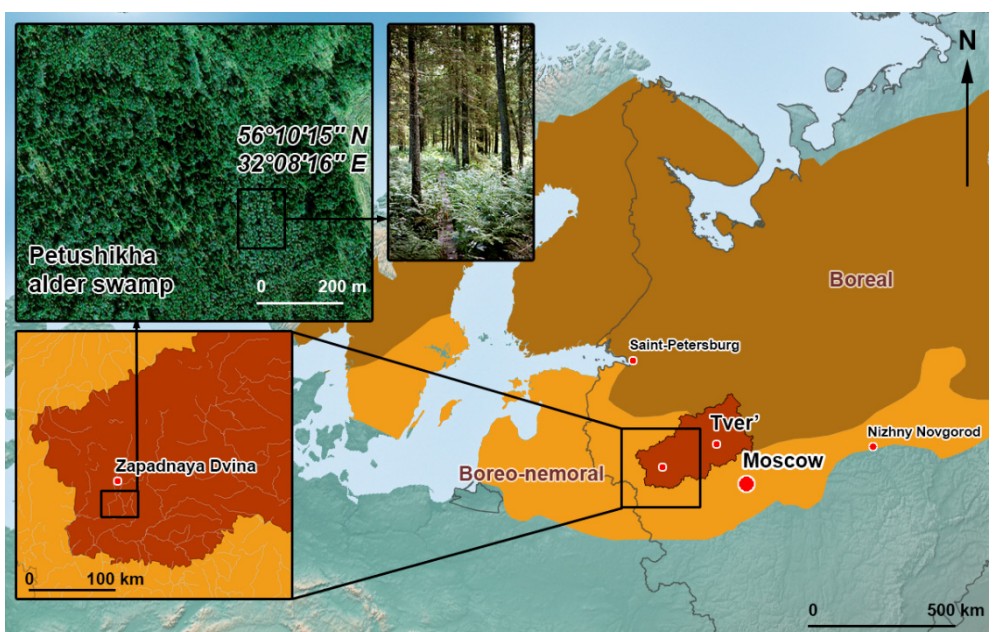

**Figure 1.** The location of the Petushikha alder swamp in European Russia according to [58].

### 2.2. Phytosociological Records

The composition of the stand is spruce-black alder forest with birch admixture; the average age of black alder is 100 years, spruce—94 years. Four layers represent the vegetation: tree, shrub, herbaceous, and mossy.

The tree layer is formed by black alder (*Alnus glutinosa* (L.) *Gaertn*), European spruce (*Picea abies* (L.) Karst.), and partly by birch (*Betula pubescens* Ehrh). The shrub layer is

dominated by bird cherry (*Padus avium* Mill.), alder buckthorn (*Frangula alnus* Mill.), gray willow (*Salix cineria* L.), bay willow (*S. pentandra* L.), black currant (*Ribus nigrum* L.), and guelder-rose (*Viburnum opulus* L.).

The background species in the herbaceous layer are lady fern (*Athyrium filix femina* (L.) Roth), meadowsweet (*Filipendula ulmaria* (L.) Maxim.), yellow loosestrife (*Lysimachia vulgaris* L.), bog arum (*Calla palustris* L.), cabbage thistle (*Cirsium oleraceum* L.), bittersweet nightshade (*Solanum dulcamara* L.), and common reed (*Phragmites australis* (Cav.). Background species of micro-elevations and tussocks are stone bramble (*Rubus saxatilis* L.), wood sorrel (*Oxalis acetosella* L.), and false lily of the valley (*Maianthemum bifolium* (L.) F.W. Schmidt). Other species are wild angelica (*Angelica sylvestris* L.), cowbane (*Cicuta virosa* L.), common nettle (*Urtica dioica* L.), marsh-marigold (*Caltha palustris* L.), marsh fern (*Thelipteris palustries* Schott.), and water horsetail (*Equisetum fluviatile* L.).

The moss cover consists of Drepanocladus exannulatus (Warnst.), D. sendtneri (Schimp. ex H.Müll.), Callurgonella Loeske, Calliergon giganteum (Schimp.), Climacium dendroides (Hedw.), Brachythecium rivulare (Schimp.), and Mnium rugicum (Laur.). Sphagnum mosses are absent. The tree layer is mainly found on the hillocks and much less in level locations, whereas in depressions it is absent.

### 2.3. Soil Cover

The soils of the black alder swamp "Petushikha" are *Fibric Histosols* (WRB), and the thickness of peat deposits underlain by loams is from 2.0 to 3.7 m. To characterize the swamp soil using a TBG-1 peat drill with a diameter of 5 cm and 50 cm nozzles, we collected samples from different soil horizons up to the parent rock. The radiocarbon (basal) age of the peat is 8.750 $\pm$ 70 calBP (IGRAS-1363). Peat dating was carried out at the Institute of Geography of the Russian Academy of Sciences; calibrated $^{14}$C dates (calBP) were calculated using the Calib 5.1 program (median) [60].

The peat had a slightly acidic reaction; the pH of the salt extract varied from 5.3 to 5.6. The ash content differed significantly: in layers of 0–30, 30–150, and 150–370 cm, it was 13–19, 10–12, and 14%–29%, respectively, which is explained by the conditions of the introduction of mineral particles. The high degree of peat decomposition (45%–50%) determines its significant (in particular, in the lower horizons) bulk density: 0.17–0.33 g/cm$^3$. The carbon content in the studied peat soil reaches 51%. Carbon stocks increase with the depth of the deposit (Table 1). According to the botanical composition, the type of peat is identified as woody eutrophic.

**Table 1.** The characteristics of peat soil in black alder swamp "Petushikha".

| Depth, cm | pH (1N KCl) | Ash Content, % | Bulk Density, g cm$^{-3}$ | Total Carbon, % | Carbon Reservoir in 10 cm Layer, kg m$^{-2}$ |
|---|---|---|---|---|---|
| 0–10 | 5.4 | 19 | 0.17 | 49 | 8 |
| 10–20 | 5.4 | 15 | 0.18 | 48 | 9 |
| 20–30 | 5.3 | 13 | 0.17 | 48 | 8 |
| 30–40 | 5.4 | 12 | 0.16 | 49 | 8 |
| 40–50 | 5.5 | 11 | 0.17 | 49 | 8 |
| 50–60 | 5.5 | 11 | 0.17 | 50 | 9 |
| 60–70 | 5.5 | 11 | 0.17 | 49 | 8 |
| 70–80 | 5.4 | 10 | 0.15 | 50 | 7 |
| 80–90 | 5.6 | 10 | 0.15 | 51 | 8 |
| 90–100 | 5.5 | 10 | 0.16 | 50 | 8 |
| 100–150 | 5.6 | 10 | 0.23 | 51 | 11 |
| 150–200 | 5.5 | 14 | 0.21 | 48 | 10 |
| 200–250 | 5.6 | 24 | 0.22 | 42 | 9 |
| 250–300 | 5.5 | 29 | 0.33 | 40 | 13 |
| 300–370 | 5.5 | 23 | 0.20 | 44 | 9 |

Note: For depths of 100–370 cm, carbon reservoirs are calculated for a 10 cm thick layer.

The degree of decomposition and the botanical composition of peat for the selected soil horizons were determined according to the protocol of [61]. Ash content was determined by calcination at a temperature of 525 °C; pH (1 N KCl) was measured on an EV-74 ion meter (Gomel Plant of Measuring Instruments—GPMI, Gomel, Belarus) with an ESL-43-07 measuring electrode [62]. The carbon content in soil samples in 1995–1998 was determined at the M.V. Lomonosov Moscow State University via dry combustion in a stream of oxygen using an AN-7529 express analyzer (GPMI, Gomel, Belarus) with a coulometric tip and in 2013–2016 on the elemental analyzer Vario MICRO cube (Elementar, Langenselbold, Germany). The bulk density of the upper horizons up to 50 cm, without disturbing the deposit, was determined using a hollow stainless-steel cylinder with a diameter of 15 cm, a height of 10 cm, a volume of 1813 cm$^3$ with two lids (upper and lower), and a cutting ring with a sharpened edge. From deeper horizons, samples were collected using a peat drill. The moisture content of the samples was determined by drying them to a constant weight at 105 °C, and then the density of peat was calculated [63].

*2.4. Field Measurements*

Methane fluxes were measured from June to October in 1995–1997, from May to September in 1998, from August to October in 2013, from July to October in 2014, and from June to October in 2015 and 2016 on four key elements of the microrelief: (1) depressions (DEP, depth −5 . . . −8 cm); (2) flat surface (FL; the average level of the flat surface was taken as the surface zero for the depth of depressions and the height of elevations); (3) micro-elevations (EL, h = 15–25 cm); (4) water surfaces (STR, streams and channels, h = −15 . . . −25 cm). On the plot under study with an area of 0.6 hectares, we carried out leveling of the soil surface in squares with a step of 2 × 2 m. This made it possible to quantitatively assess the representativeness of various microsites. Depressions occupy 10% of the area, flat surfaces—35%, micro-elevations (including tussocks)—20%, and water surfaces—3%. The rest of the surface (32%) is represented by near-steam tussocks, pierced by large tree roots, and methane fluxes were not measured here (due to the inability to install methane chambers hermetically).

The measurements of CH$_4$ fluxes were carried out using static chambers installed at each microsite (DEP, FL, EL, and STR) in 3–4 spatial replicates (a detailed description of the frequency of measurements can be found in results section, Table 2). The chambers were light-proof white cylinders made of low-pressure high-density polyethylene (HDPE) with a volume of 32.5 L, a height of 55.9 cm, and a base area of 0.11 m$^2$. The chambers were installed on metal bases with a water seal which were cut beforehand into peat soil of 10 cm and left in there for the entire period of the study. The vegetation under the chamber remained untouched. On water surfaces, the storage chambers were mounted onto Styrofoam floats serving as bases. To prevent the squeezing of gases from the soil, each microsite was equipped with wooden gangways used for gas sampling from the chambers. The measurements were started in the first half of the day. All chambers were equipped with compensatory containers which made it possible to carry out gas sampling without changing the atmospheric pressure inside them. The exposure time was 24 h. Gas mixing in the chamber and its subsequent collection was carried out using a gas sampling device through a fitting into pre-evacuated sealed PVC package equipped with nipple devices. In the laboratory, gas samples were collected through the nipple devices with a syringe in triplicate for further analysis. The analysis of the methane concentration was carried out in a laboratory no later than 3 h after gas sampling.

In 1995–1998, the CH$_4$ concentration was measured on a Chrom-5 gas chromatograph (Laboratory instruments, Prague, Czech Republic) equipped with two flame ionization detectors (carrier gas—nitrogen, sorbent—Porapak Q). In 2013–2016, the measurements were carried out on a Kristall-2000 M gas chromatograph (Chromatec, Yoshkar-Ola, Russia) equipped with two flame ionization detectors (carrier gas—helium, sorbent—Porapak Q). Each gas sample was analyzed in triplicate. Calibration of the chromatographs was carried out before and after the analysis of the entire batch of gas samples. For calibration, we

used a gas mixture containing 22.2, 40.1, and 84 ppm methane in nitrogen. The calibration mixture of gases was prepared in the laboratory of soil microbiology of the Institute of Microbiology of the Russian Academy of Sciences.

$CH_4$ flux (mgC m$^{-2}$ h$^{-1}$) was calculated using the following formula:

$$F = \frac{\frac{dC}{dt} \times V \times M}{S \times Vm}, \tag{1}$$

where $C$ is the $CH_4$ concentration (ppm C-$CH_4$), $t$—time (h), $V$ is the chamber volume (m$^3$), $M$ is the molar mass of $C$ (g/mol), $V_m$ is the molar volume of $CH_4$ (l/mol), and $S$ is the area of the horizontal section of the chamber (m$^2$).

We measured the temperature of the air ($T_{air}$) and soil ($T_{soil}$) at a depth of 50 cm, as well as the level of GWL from the soil surface in parallel with the determination of $CH_4$ emissions. Negative GWL values indicated the water level below the soil surface, while positive values indicated the water level was above it. The air and peat temperatures were measured with sensors included among the Li-COR accessories. GWL measurements were carried out in observation wells using a hollow aluminum tube 1 cm in diameter and 150 cm in length with 1 mm notching on the outside. These wells were hollow plastic pipes 3 cm in diameter perforated along their entire length, one of the ends of which was closed with a wooden plug; the lower ends of the pipes reached the mineral bottom of the underlying peat deposit. In addition, 3 rain gauges were placed on the soil surface in the black alder forest. The precipitation was measured with a measuring glass with 100 divisions, each of which corresponded to 0.1 mm of precipitation.

The weighted average total soil methane fluxes in the study area were assessed based on the data on the representativeness of various microsites. The calculation was carried out according to the following formula (according to previous study [64]):

$$F_D = \sum_{i=1}^{n} D_i \times F_i, \tag{2}$$

where $F_D$ is the weighted average flux, mgC m$^{-2}$ h$^{-1}$, $n$ is the number of types of microsites, $i$ is the number of the microsite, $D$ is the share of the area occupied by the microsite, $F$ is the methane flux at the microsite (mgC m$^{-2}$ h$^{-1}$).

### 2.5. Meteorological Conditions

The amount of precipitation and air temperature (annual and average for the period from 1993 to 2020) calculated based on data from weather stations ("Toropets", "Velikiye Luki" and "Smolensk" [65]) are presented in Figure 2.

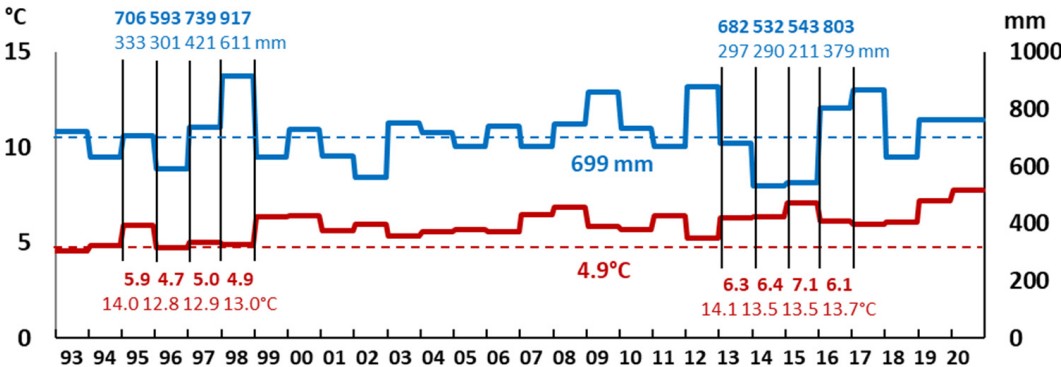

**Figure 2.** The average annual (for 1995–1998; 2013–2016), average for 1993–2020 (dotted line) air temperature (bottom) and precipitation (top), based on data from the weather stations "Toropets", "Veliky Luki" and "Smolensk" for the area, in which the black alder forest "Petushikha" is located.

The average annual air temperature and amount of precipitation from 1993 to 2012 were 4.9 °C and 699 mm, and for the summer months (from June to October, J-O)–13.6 °C and 375 mm, respectively. The annual amount of precipitation varied greatly in individual years within two periods (1995–1998 and 2013–2016): from 532 mm (290 mm for the J–O period) of precipitation in the dry 2014 year to 917 mm (611 mm for the J-O period) of precipitation in the extremely humid year 1998. It should be noted that the most significant variation is typical for summer periods: from year to year, the amount of precipitation can differ by 2–3 times and in some cases (1998) can almost reach the annual norm. At the same time, the average annual precipitation did not change from year to year by more than 1.5 times. The average annual air temperature changed from year to year to a lesser extent, while maintaining an upward trend: from 4.9 °C in 1998 (for the period from 1995 to 1998, it was on average 5.1 °C) to 7.1 °C in 2015 (for the period from 2013 to 2016, it was, on average, 6.5 °C).

### 2.6. Data Analysis

Descriptive statistics of field measurements (medians, average, min, max, IQR, and n) was provided for measured fluxes. Statistic difference between medians of fluxes on different microsites was checked using the Kruskal–Wallis test ($n = 293$ for each of 4 groups, threshold value $p = 0.05$; with recourse to nonparametric statistics since the fluxes values were not normally distributed), for pairwise comparison. The probability density of methane fluxes at individual sites was calculated using the "ksdensity" function (Matlab).

## 3. Results

### 3.1. Methane Fluxes

An analysis of the probability density distribution of $CH_4$ fluxes (mgC m$^{-2}$ h$^{-1}$) obtained in the summer–autumn periods of 1995–1998 and 2013–2016 for 4 sites (EL, FL, DEP, STR) is presented in Figure 3.

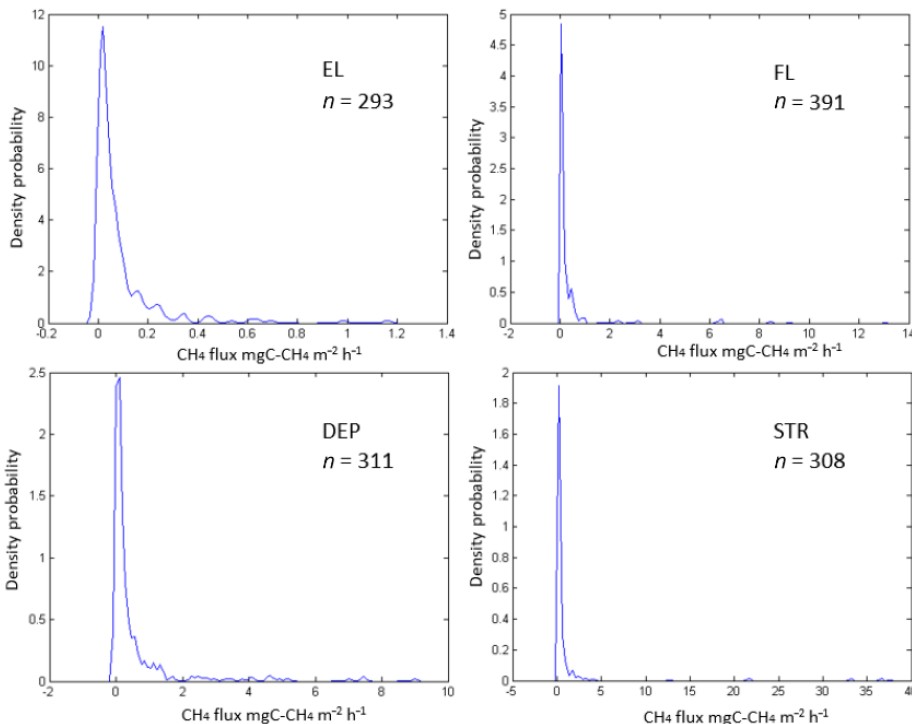

**Figure 3.** The probability density distribution of $CH_4$ fluxes at four microsites: elevations (EL), flat surface (FL), depression (DEP), stream (STR). All data from the periods 1995–1998 to 2013–2016 were used.

The obtained distributions have a log-normal (DEP site: Kolmogorov–Smirnov (KS) and Shapiro–Wilk (SW) test for normality of natural logarithm of methane fluxes show $p > 0.20$ and 0.13, respectively) or similar (FL, STR site: KS test show $p > 0.20$ excluding upper and lower 10% of natural logarithm of methane fluxes) form for relief elements (excluding EL site), which confirms a well-known statement about the abnormal distribution of methane fluxes in swamps and supports the need to use nonparametric methods to describe the results obtained.

The results of statistical generalization of all obtained methane fluxes for 4 sites are shown in Figure 4. According to the Kruskal–Wallace test (total $n = 1172$, $p < 0.01$), medians of fluxes differ significantly for relief elements (sites). The smallest fluxes (1Q-median-3Q, mgC m$^{-2}$ h$^{-1}$) are typical for elevations (0.01–0.03–0.09; EL), somewhat larger for a flat surface (0.02–0.06–0.19; FL), and the largest for depressions (0.04–0.14–0.43; DEP) and water surfaces (0.10–0.21–0.44; STR).

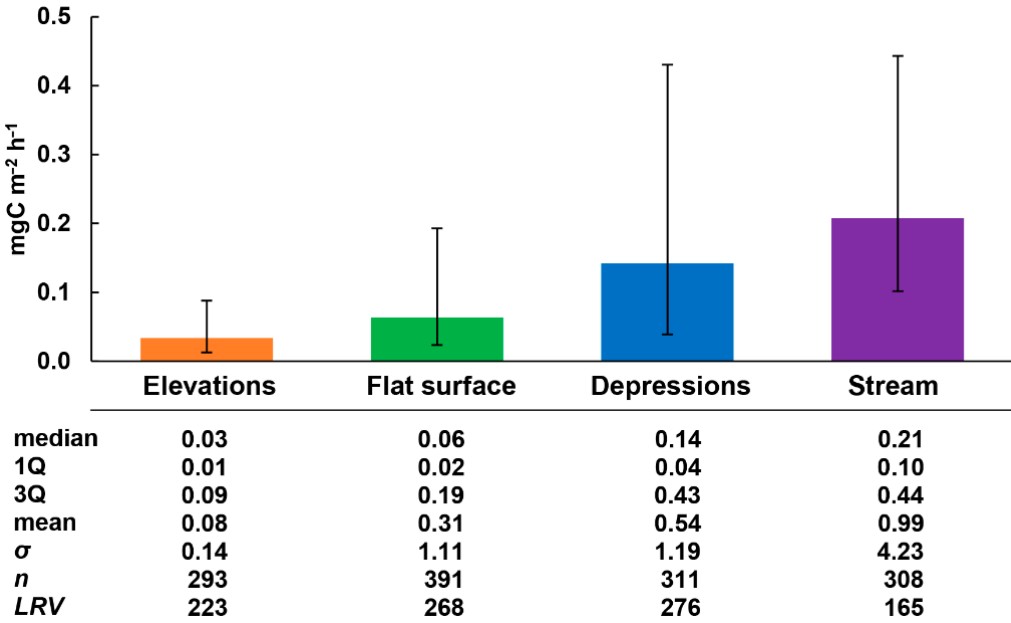

|  | Elevations | Flat surface | Depressions | Stream |
|---|---|---|---|---|
| median | 0.03 | 0.06 | 0.14 | 0.21 |
| 1Q | 0.01 | 0.02 | 0.04 | 0.10 |
| 3Q | 0.09 | 0.19 | 0.43 | 0.44 |
| mean | 0.08 | 0.31 | 0.54 | 0.99 |
| $\sigma$ | 0.14 | 1.11 | 1.19 | 4.23 |
| *n* | 293 | 391 | 311 | 308 |
| *LRV* | 223 | 268 | 276 | 165 |

**Figure 4.** Medians (bars), 1Q and 3Q (lower and upper whiskers, respectively) of measured CH$_4$ fluxes (mgC m$^{-2}$ h$^{-1}$), and their basic statistics at 4 sites in alder swamp Petushikha. Statistic difference between medians was checked using the Kruskal–Wallace test (total $n = 1172$): $p < 0.01$ for all cases. The level of relative variability (LRV) is defined as one half of interquartile range divided by median of methane flux sample and multiplied by 100% (a nonparametric analog of coefficient of variation).

The lower and upper whiskers shown in the diagram and corresponding to 1Q and 3Q, respectively, illustrate the total absolute variability of methane fluxes within the limits of seasonal and interannual variability. The "amplitude" of absolute flux variability is quite large: IQR (the interquartile range) at all sites exceeds the median and comprises 0.08, 0.17, 0.39, and 0.34 mgC m$^{-2}$ h$^{-1}$ in the EL-FL-DEP-STR sequence, respectively. The upper bounds on the uncertainty (3Q) of the fluxes at the DEP and STR sites are practically the same, and the lower bounds (1Q) are close at the DEP, FL, and EL sites, while differing significantly from the STR site. The level of relative flux variability (LRV, %) was chosen as a non-parametric analogue of the coefficient of variation [66] and calculated by the formula:

$$LRV = \frac{IQR}{2 \times F_m} \times 100, \tag{3}$$

where *LRV*—the relative flux variability in %; *IQR*—the interquartile range (mgC m$^{-2}$ h$^{-1}$); $F_m$—the median flux (mgC m$^{-2}$ h$^{-1}$). The level of relative flux variability (LRV) increases in the STR-El-FL-DEP sequence and comprises 165, 223, 268, and 276%, respectively. The

smallest LRV is characteristic of STR: it turned out to be less than that observed at DEP, which is the site closest in terms of moistening conditions. At the FL and DEP sites, the relative variability is almost identical and differs slightly from that at the EL site.

Thus, generalizing the obtained results, we can conditionally single out the following groups among the studied sites: (i) "large" methane flux (average value of 0.18 mgC $m^{-2}$ $h^{-1}$)—DEP, STR; "small" flux (average value of 0.05 mgC $m^{-2}$ $h^{-1}$)—EL, FL; (ii) "large" absolute variability (*IQR*) (average value of 0.37 mgC $m^{-2}$ $h^{-1}$)—DEP, STR; "small" absolute variability (average value of 0.13 mgC $m^{-2}$ $h^{-1}$)—EL, FL; (iii) "large" relative variability (LRV) (average value of 256%)—EL, FL, DEP; "small" relative variability (165%)—STR.

### 3.2. Spatial Variability of Methane Fluxes

To assess the spatial variability of methane fluxes at the sites (EL, FL, DEP, STR), it was necessary to minimize the temporal variability (in terms of this work, it means seasonal and interannual, since the daily variability was not assessed). To do this, we calculated the level of relative spatial variability (spLRV—spatial LRV) in a sample of fluxes measured at all 4 sites under consideration on the same day. The results of an assessment of the spatial variability of methane fluxes are given in Table 2. The colors indicate the following spLRV gradation: green, ≤100%, orange, 100%–200%, dark orange, >200%.

It should be noted that on the majority of separately considered days (76 out of 105), when methane fluxes were measured, their variability between different sites did not exceed 100%. Average values of spLRV over the years vary from 73% in 1996 and 2015 up to 193% in 2016. In addition to the large spLRV in 2016, when measurements were made during only 5 field campaigns, significant spatial variability of fluxes was noted in 1995 and 1997 (119 and 95%, respectively). In June and July 1995, as well as in July and September 1997, the spatial variability of fluxes in some cases reached 200%–400% and even up to 800%. The average spLRV over the years was 105%, which indicates a significant spatial heterogeneity of methane fluxes.

**Table 2.** The spLRV (level of relative variation, %) of methane fluxes in space, calculated between all sites (El, FL, DEP, STR) within individual measurement days. The colors show the LRV gradation: green—≤100%, orange 100%–200%, dark orange—>200%.

| spLRV, % | | | | | | | | | | | | | | | |
|---|---|---|---|---|---|---|---|---|---|---|---|---|---|---|---|
| Date | | | | | | | | | | | | | | | |
| 1995 | | 1996 | | 1997 | | 1998 | | 2013 | | 2014 | | 2015 | | 2016 | |
| 12 June | 96 | 31 May | 112 | 4 June | 115 | 9 May | 580 | 27 Aug | 18 | 9 July | 87 | 16 June | 137 | 24 June | 164 |
| 13 June | 197 | 5 June | 57 | 11 June | 97 | 13 May | 40 | 5 September | 39 | 18 July | 115 | 23 June | 85 | 12 July | 212 |
| 14 June | 236 | 12 June | 50 | 20 June | 78 | 22 May | 72 | 20 September | 58 | 25 July | 80 | 9 July | 43 | 30 Aug | 413 |
| 24 June | 60 | 19 June | 36 | 25 June | 95 | 27 May | 63 | 8 October | 65 | 5 Aug | 167 | 21 July | 51 | 15 September | 89 |
| 26 June | 72 | 26 June | 74 | 2 July | 195 | 3 June | 53 | | | 22 Aug | 30 | 6 Aug | 68 | 6 October | 89 |
| 28 June | 142 | 2 July | 75 | 9 July | 225 | 13 June | 78 | | | 4 September | 17 | 10 September | 110 | | |
| 30 June | 410 | 8 July | 76 | 16 July | 74 | 18 June | 60 | | | 9 September | 35 | 29 September | 49 | | |
| 14 July | 118 | 26 July | 88 | 23 July | 810 | 24 June | 57 | | | 8 October | 66 | 6 October | 42 | | |
| 15 July | 150 | 1 August | 37 | 30 July | 58 | 2 July | 49 | | | | | | | | |
| 16 July | 161 | 7 August | 30 | 5 August | 61 | 8 July | 61 | | | | | | | | |
| 22 July | 138 | 14 August | 72 | 15 August | 51 | 15 July | 37 | | | | | | | | |
| 28 July | 53 | 21 August | 67 | 20 August | 36 | 23 July | 19 | | | | | | | | |
| 2 August | 46 | 29 August | 54 | 27 August | 34 | 13 August | 44 | | | | | | | | |
| 8 August | 47 | 4 September | 179 | 4 September | 35 | 19 August | 35 | | | | | | | | |
| 29 August | 79 | 11 September | 54 | 10 September | 109 | 28 August | 46 | | | | | | | | |
| 6 September | 106 | 18 September | 81 | 18 September | 105 | 3 September | 74 | | | | | | | | |
| 14 September | 69 | 25 September | 70 | 25 September | 128 | 9 September | 47 | | | | | | | | |
| 18 September | 73 | 2 October | 76 | 3 October | 132 | 17 September | 30 | | | | | | | | |
| 27 September | 176 | 10 October | 91 | 8 October | 59 | 23 September | 82 | | | | | | | | |
| 5 October | 88 | 15 October | 75 | | | | | | | | | | | | |
| 11 October | 31 | | | | | | | | | | | | | | |
| 18 October | 62 | | | | | | | | | | | | | | |
| 24 October | 114 | | | | | | | | | | | | | | |
| **1995** | | **1996** | | **1997** | | **1998** | | **2013** | | **2014** | | **2015** | | **2016** | |
| **Mean** | 119 | 73 | | 95 | | 131 / 53 * | | 80 | | 75 | | 73 | | 193 | |
| **Total mean** | | | | | | | | 105 | | | | | | | |

Note: * median of spatial LRV.

1998 is also characterized by a large average spLRV, but the reason for this is the methane flux measurements made in May (spLRV is equal to 580%): when calculating the median (instead of the average) of the annual spLRV variability, we obtained 53% for 1998, which characterizes the spatial variability in this year as the smallest among others.

### 3.3. Seasonal Variability of Methane Fluxes

Seasonal variability of methane fluxes over the years at the sites is shown in Figure 5. It is rather difficult to identify a single characteristic pattern of seasonal dynamics of methane fluxes for all sites. In 1995–1997, the relatively drained EL (Figure 5, orange) and FL (green) sites were characterized by an increase of methane fluxes from June to August and then a decrease towards September (a "bell" pattern).

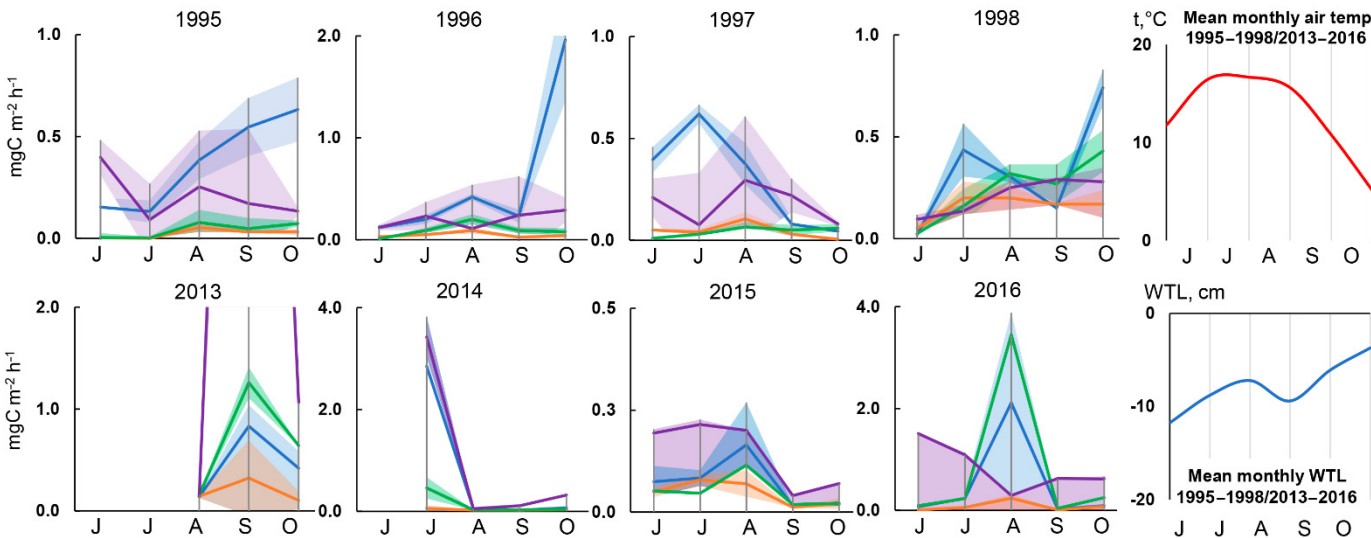

**Figure 5.** The medians ± IQR of measured $CH_4$ fluxes (mgC m$^{-2}$ h$^{-1}$) during the season on EL (orange), FL (green), DEP (blue) and STR (purple) at 4 sites in alder swamp "Petushikha". At the STR site, peak methane flux in September 2013 was reached: 11.7 mgC m$^{-2}$ h$^{-1}$.

The same dynamics were typical for the EL, FL, and DEP sites in 2015. At the STR site (Figure 5, purple), the dynamics were rather the opposite: in 1995–1997, 2014, and 2016, there was a decrease in the magnitude of methane fluxes from June to July (1995, 1997) or from July to August (1996, 2014 and 2016), followed by an increase by August or September (a "bowl" pattern). At the DEP sites, the character of seasonal dynamics in some cases had the "bell" pattern (1997, 2013, 2016), in others—the "bowl" pattern (2014). In 1995, 1996 and 1998, methane fluxes at the DEP sites were almost constantly growing from June to October, with a slight decrease in September.

As a quantitative measure of the seasonal variability (sLRV—seasonal LRV) of methane fluxes at the sites under consideration (EL, FL, DEP, STR), we used the level of relative variability for fluxes measured at the same sites during the season (Table 3). To exclude the effect of interannual variability, different years were considered separately.

**Table 3.** The sLRV (level of relative variation, %) of methane fluxes during the season, calculated for all sites (EL, FL, DEP, STR) within individual measurement years. The colors show the sLRV gradation: green—≤100%, orange 100%–200%, dark orange—>200%.

| | | | | | sLRV, % | | | | | |
|---|---|---|---|---|---|---|---|---|---|---|
| Site/Year | 1995 | 1996 | 1997 | 1998 | 2013 | 2014 | 2015 | 2016 | Mean | Total Mean |
| EL | 767 | 94 | 78 | 76 | 101 | 68 | 64 | 98 | 168 | |
| FL | 183 | 78 | 97 | 91 | 40 | 216 | 57 | 125 | 111 | 144 |
| DEP | 78 | 99 | 131 | 115 | 48 | 777 | 95 | 180 | 190 | |
| STR | 74 | 54 | 108 | 45 | 139 | 294 | 79 | 45 | 105 | |
| Mean | 275 | 81 | 104 | 82 | 82 | 338 | 74 | 112 | | |

As with spatial variability, in some cases sLRV exceeded 100% (12 out of 32 seasons). The average (by sites) relative seasonal flux variability increases in the STR-FL-EL-DEP sequence and comprises 105, 111, 168, and 190%, respectively. The average sLRV was 144%, which exceeds the spatial variability (spLRV) by almost 1.5 times. Water and flat surfaces (STR and FL, respectively) were relatively more stable sources of methane during the summer seasons of all years under consideration, while the elevations (EL) and depressions (DEP) were the least stable sources. The highest relative seasonal variability of methane fluxes was observed in 2014 at almost all sites (216%–777%), except for EL (68%).

### 3.4. Interannual Variability of Methane Fluxes

To assess the interannual variability of methane fluxes, we calculated the variability of their monthly aLRV medians (annual LRV) divided by sites. The results obtained are presented in Table 4. The measurements performed in different months were replicated over the years to varying degrees: for example, in May the results were obtained only for 2 years, and in other months for at least 6 years. It is very likely that the aLRV assessment for May is the least representative of all the months.

**Table 4.** Interannual monthly variability of methane fluxes aLRV (level of relative annual variation, %) for 8 years of observations (1995–1998 and 2013–2016). Calculated with division by microrelief elements (sites EL, FL, DEP, STR). The colors show the aLRV gradation: green—≤50%, orange 50%–100%, dark orange—>100%.

| Site/Month | aLRV, % | | | | | | |
|---|---|---|---|---|---|---|---|
| | EL | FL | DEP | STR | Mean | *n* (Years) | Years |
| May | 34 | 39 | 32 | 1 | 27 | 2 | 1996,1998 |
| June | 43 | 118 | 91 | 50 | 76 | 6 | 1995–1998, 2015,2016 |
| July | 20 | 132 | 63 | 112 | 82 | 7 | 1995–1998, 2014–2016 |
| August | 43 | 53 | 46 | 35 | 44 | 8 | 1995–1998, 2013–2016 |
| September | 82 | 141 | 183 | 46 | 113 | 8 | 1995–1998, 2013–2016 |
| October | 52 | 77 | 262 | 63 | 113 | 7 | 1995–1997, 2013–2016 |
| Mean | 46 | 93 | 113 | 51 | | | |
| Total | | | 75 | | | | |

The average interannual variability increases for sites in the EL-STR-FL-DEP sequence and comprises 46, 51, 93 and 113%, respectively. It should be noted that the highest interannual variability was characteristic of the DEP sites, where the intraseasonal variability (see Section 3.3) was also the highest among other relief elements.

On average, September and October were the most variable (113%) in terms of methane emissions over the years, while May and August were the least variable (27%–44%). However, upon a detailed examination of individual elements of the microrelief, it can be noted that this rule does not apply to all sites. For example, at the elevations (EL), the most variable months were indeed September and August, but the smallest aLRV is typical for July. At the flat surface (FL) sites, almost all summer months were very variable: aLRV was 1.5–2 times higher than the average value at other sites, and only in August did it follow the general downward trend. At depressions (DEP), methane fluxes in September and October were maximally variable over the years (including in comparison with other sites), along with June and July (which was also typical for FL). In different years in August, the fluxes at the DEP site, as well as at other sites, remained very stable. Finally, among all sites, the water surface sites (STR) were characterized by the most stable methane fluxes during all months (especially in May), except for July.

## 4. Discussion

### 4.1. CH₄ Fluxes

The results of measurements of methane fluxes in black alder swamp "Petushikha" are consistent with previous estimates for similar (boreal and temperate) and ecologically

close (tropical) habitat types (Table 5). For example, in birch and spruce swamps in Western Siberia [56,57], methane fluxes varied in the range from 0.02 to 0.6 mgC m$^{-2}$ h$^{-1}$, the lower limit of which almost completely coincides with the measurements at the EL site (1Q = 0.01). However, its upper limit exceeds the fluxes at the STR site even considering the uncertainty we obtained (3Q = 0.44 mgC m$^{-2}$ h$^{-1}$).

**Table 5.** Literature data on the magnitude of methane fluxes in swamps in various regions of the World (if the initial dimensions differed from those presented in the table, the fluxes were recalculated in mgC m$^{-2}$ h$^{-1}$ by directly dividing by the number of days in a year and hours in a day).

| Ecosystem Type | Location | CH$_4$, Flux mgC m$^{-2}$ h$^{-1}$ | Reference |
|---|---|---|---|
| Temperate Swamp | Canada | −0.1–0.8 | [18] |
| Spruce swamp | Norway | 0.05 | [20] |
| Spruce forest | Finland | −0.02–3.7 | [23] |
| Alder swamp | Canada | 0.01–0.02 | [45] |
| Birch and spruce swamp | Russia (West Siberia) | 0.02 (1Q-0.03; 3Q-0.36) | [56] |
| | | 0.05–0.6 | [57] |
| Birch and hemlock swamp | Canada | 0.1–0.4 | [67] |
| Needle-leaved Swamp Broad-leaved Swamp | Canada and USA | 0.13–0.99 3.9–6.7 | [68] |
| Birch and larch swamp | China | −0.03–0.10 | [69] |
| Acer and wild olive swamp | USA (Virginia) | −0.02–0.6 | [70] |
| Tropical swamp | Malaysia | 0.9–1.3 | [71] |
| Acer and wild olive swamp | USA (Virginia) | 0–2.3 | [72] |
| Tamarack and cedar swamp | Canada | 0-3.8 | [73] |
| Swamp | Pan-arctic | 0.31–0.37 (mean) 0.14 (median) | [74] |
| Alder swamp | European Russia | EL site, 1Q-median-3Q: 0.01-0.03-0.09 FL: 0.02-0.06-0.19 DEP: 0.04-0.14-0.43 STR: 0.10-0.21-0.44 | This study |

A closer example in terms of the range of observed methane fluxes are registered in a birch and hemlock swamp in Canada [67] and birch and spruce swamps in Russia (West Siberia) [57], as well as at a spruce swamp in Norway [20]. In the first case, the values of fluxes registered in the course of the current work at the EL, FL, DEP, and STR sites almost completely coincided (considering the uncertainty limits) with the presented range: 0.1–0.4 mgC m$^{-2}$ h$^{-1}$ [57,67]. In the second case, the fluxes that we registered at the EL (1Q–3Q: 0.01–0.09), FL (0.02–0.19) and DEP (0.04–0.43) sites, considering the uncertainty, did not differ from the data of [20] (0.05 mgC m$^{-2}$ h$^{-1}$).

The values of methane fluxes in similar ecosystems under similar climatic conditions (Finland, Canada, and Northeast China) also generally do not differ from those obtained by us in European Russia [45,68]. However, some works [18,23,69] reported methane assimilation at a rate of up to −0.1 mgC m$^{-2}$ h$^{-1}$, which was not observed at all in the black alder swamp we studied.

The range of the total variability of methane fluxes was 2 orders of magnitude in several cases [23,68,73], while the same indicator in the "Petushikha" swamp hardly reached 1 order of magnitude. Both greater methane uptake and greater dispersion of flux values can be achieved by including the sites characterized by boundary conditions into consideration:

it is very likely that the more drained forests surrounding the "Petushikha" black alder swamp may differ in methane uptake, while the more moistened sites located closer to the edge of the adjacent wetland may have higher $CH_4$ emissions. The issue of drawing discrete boundaries between continuous ecosystems to adequately assess the characteristic methane fluxes in them is of great importance. We believe that classification of the waterlogged forests should be based primarily on the characteristics of the vegetation cover (tree, shrub, and herbaceous layers) as an indicator of long-term prevailing moisture conditions in the ecosystem under consideration; it is one of the key factors determining the formation of anaerobic conditions and serves as an important driver of methanogenesis.

The obtained values of methane fluxes are very typical for the swamps of the panarctic region (the flux median is 0.14 mgC m$^{-2}$ h$^{-1}$—[74]), slightly exceeding the latter, likely due to more favorable climatic conditions of the boreo-nemoral zone (some swamps used to calculate the methane flux typical for the Arctic in the [74] are located close to or beyond the Arctic Circle). At the same time, the southern swamps of USA Great Dismal Swamp, Virginia—[68,70,72], Malaysia [71], and other tropical regions that are not considered here, are characterized by significantly higher methane emissions than their counterparts in the boreo-nemoral and boreal zones. This is very likely a natural consequence of an increase in the favorability of climatic conditions, the abundance of mineral nutrition elements, the moistening conditions of tropical forests, etc.

Thus, the current results generally agree well with other studies. However, at the same time it was possible to demonstrate the differences in the magnitude of methane fluxes due to the microrelief of the area, and, as a result, different typical WTLs. In addition, our estimates include seasonal and interannual variability and, very importantly, are presented as a probability density of the methane flux.

### 4.2. Variability of $CH_4$ Fluxes

The average variability of methane fluxes increases in the series: interannual (by month—aLRV) < spatial (spLRV) < seasonal (sLRV) and amounts to 75, 105 and 144%, respectively. This rule is also preserved when comparing the interannual and seasonal flux variability at individual sites: 46 < 168 (aLRV < sLRV at EL), 93 < 111 (FL), 113 < 190 (DEP), 51 < 105 (STR). However, the difference between the interannual and seasonal variability is 3.5 times at the most drained EL, but it is no more than 1.2–2.1 times on the more wet FL, DEP, and STR. So far, few studies have been published that allow us to assess the spatio-temporal variability of methane fluxes in waterlogged forests; they are listed in Table 6.

**Table 6.** Literature data on the magnitude of various types of variability of methane fluxes in Swamps in various regions of the world (data from the figures were used to calculate the LRVs in some cases).

| Type of Variability | LRV, % | n | Notes | Reference |
|---|---|---|---|---|
| Seasonal (Site 6) | 92 | 11 | 1986 | |
| | 156 | 32 | 1987 | |
| | 350 | 17 | 1988 | |
| Seasonal (Site 7) | 420 | 14 | 1987 | |
| | 163 | 11 | 1988 | |
| Annual (Site 6) | 41 | 10 | April | [18] |
| | 44 | 15 | July | |
| | 35 | 10 | August | |
| | 250 | 10 | September | |
| | 36 | 10 | October | |
| Annual (Site 7) | 55 | 6 | May | |
| | 300 | 6 | June | |
| | 50 | 6 | July | |
| Spatial | 63 | 40 | 1987 | |
| | 103 | 32 | 1988 | |

**Table 6.** *Cont.*

| Type of Variability | LRV, % | n | Notes | Reference |
|:---:|:---:|:---:|:---:|:---:|
| Seasonal | 200<br>213 | 6<br>8 | Site: PR1<br>PR2 | [20] |
| Spatial | 110<br>44 | 17<br>12 | Site: fp_6(7-9)<br>fp_5(8) | [30] |
| Spatial | 45<br>40<br>246 | 6<br>9<br>14 | Site: Tr.PWF_2<br>Tr.WF/RB_2.1(2)<br>Site: fp_2(3,4) | [57] |
| Seasonal | 425<br>325 | 12<br>12 | Birch swamp<br>Larch swamp | [69] |
| Seasonal | 200 | 21 | March–November | [70] |
| Spatial | 364 | 9 | Table 1 | [72] |

Somewhat bigger—sLRV 200% (however comparable with the results of our study) seasonal variation of the methane fluxes was found in [70]. A similar result of the sLRV was obtained during one season in a Great Dismal Swamp located in a humid subtropical climate: to calculate the seasonal variability of methane fluxes, the period from March to November inclusive was used.

The authors examined the results of 3-year methane flux measurements at two spatially distinct sites in [18], which allows for calculating the corresponding LRVs. In general, the average seasonal variability comprised 200%–300% at different sites in this work, which distinguishes it as the most significant one among others: on average, the spatial variability comprised 83% there, while the interannual variability was 101%, which gives more weight to interannual variability in contrast to our study. According to this work [18], the average variability of methane fluxes increases in the series: spatial (spLRV) < interannual (by month—aLRV) < seasonal (sLRV). As in the current study, the seasonal variability here was the greatest, but the interannual variability exceeded the spatial one. Perhaps this can be explained by the fact that the measurements were carried out only for two years and only at two spatially different sites.

Extremely high (comparing to our study and [18]) seasonal variability was observed in [69], where it was 375%; however, there are opposite examples: seasonal variability did not exceed 13% in [20], although observations were made only within one season in both cases. It should be noted that the measurement sites were heavily drained (the water level never rose above 40 cm below the soil surface) in the first case [69] and such a large seasonal variability in methane fluxes was almost completely due to temperature variability ($R^2$ was up to 0.895 between methane flux and soil temperature at 5 cm depth). In the second case, the sites studied were wetter, and the water level fluctuated below the soil surface by about 20–10 cm during the season. It is possible that the faster and greater temperature variability at drained sites during the season leads to an increase in seasonal variability in methane fluxes compared to wetter sites, which are characterized by the absence of sharp changes in water level during the season. This assumption is partly confirmed by our results: as noted above, the greatest seasonal variability of methane fluxes was noted on the EL sites (driest) and the smallest one in the STR sites (wettest) (Table 3).

As our earlier studies were also limited to only one year of observations, they can only be interpreted in terms of calculating the spatial variability of methane fluxes: on average, spLRV comprised here 77%–110% [30,57], which is also close to the results of the present work. It should be noted that in [57], spatially heterogeneous (WTL ranged from 30 cm below the soil surface up to 7 cm above it) sites (fp_2-4) characterized by more variable methane fluxes (spLRV = 246%) than more homogeneous (WTL from 25 to 0 cm) ones (Tr.PWF_2 and Tr.WF/RB_2.1(2)), where spLRV was only about 40%–45%. A similar relationship was noted in [30], where spLRV of spatially heterogeneous sites (WTL from 75 up to 5 cm) was bigger (110%) than spatially homogenous (44%).

Finally, extremely high (comparing to our and all other studies—Table 6) spatial variability was observed in (Table 1 in [72]), where it was 364%. The key discovered driver of methane emission variability here is soil moisture and water table level. In addition, the sites used for the spatial variability of methane fluxes calculating in this study differed in the composition of the vegetation cover.

Thus, we were able to compare the variability of methane fluxes in space, during the season, and over different years based on long-term monitoring, which gives us an idea of the importance each of the sources of variability possesses in comparison with others.

### 4.3. Sources of Spatial Variability of CH_4 Fluxes

The spatial variability of methane fluxes in different areas we studied is a consequence of the heterogeneity of microrelief conditions and, therefore, the typical water table level. Figure 6 shows the relationship between median methane fluxes at spatially different sites and median water table calculated based on data for the entire observation period. The errors in both cases are presented as an interquartile range. It should be noted that in the case of methane fluxes, the error range includes all considered types of temporal variability: intraseasonal and interannual. Nevertheless, despite the significant scatter of the flux values observed over 8 seasons, there is an obvious tendency for the increase in the value of methane emissions as the WTL increases. Undoubtedly, this trend will change with a further increase in the GWP, which is noticeable to some extent due to the small difference between the DEP and STR sites. However, it is obvious that WTL is an extremely important factor in the variability of fluxes in space.

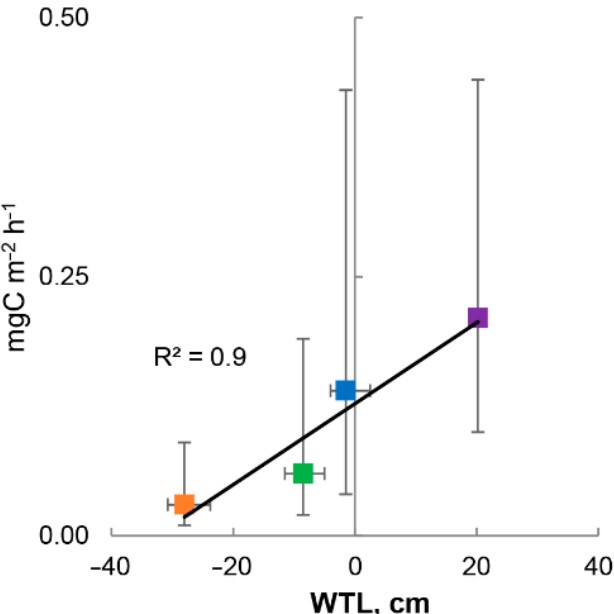

**Figure 6.** Linear regression between medians ± IQR of measured CH_4 fluxes (mgC m$^{-2}$ h$^{-1}$) on EL (orange), FL (green), DEP (blue) and STR (purple) sites and medians ± IQR of water table levels in alder swamp "Petushikha".

WTL became the key factor in the spatial variability of methane fluxes, and similar results have been obtained earlier in numerous works [18,65,73]. The water table is a major driver for dividing the soil profile into aerobic and anaerobic zones, which directly affects the net methane flux, which is the sum of its production, oxidation, and transport [75,76]. The dynamics of water table level is largely determined by the amount of precipitation and total evapotranspiration, as one of the key factors of methanogenesis, under unchanged soil conditions [77,78]. However, it turned out to be difficult to identify statistically significant factors of methane flux variability over time (during a season or over different years of ob-

servations). So, we decided to additionally check whether the amount of precipitation and the average air temperature in summer influenced the spatial differences in methane fluxes. It was observed that there was record precipitation in 1998 among 8 years of observations, both for the year (917 mm) and for the summer period (from June to October—611 mm). It is likely that high soil moisture and an increase in WTL (water table level) could lead to a "smoothing out" of differences in methane fluxes between sites with a simultaneous increase in their value. Indeed, the second largest for the period of observations median of methane emissions per season was observed in 1998.

Table 7 presents the medians of fluxes (mgC m$^{-2}$ h$^{-1}$), the sums of annual precipitation, and the average summer air temperatures for all sites in the chronological order from 1998 to 2016. The largest methane fluxes were observed in a very wet and cool 1998, and in a relatively dry and hot 2013, which was surprising. Methane fluxes and their spatial variability can increase (2013) with an increase in air temperature in summer and with the preservation of average moistening conditions; on the contrary, it can dramatically fall (2014 and 2015) with an average temperature in summer and a decrease in precipitation. The smallest methane fluxes over the observation period were registered in 2014 and 2015, which were the driest in 8 years and at the same time quite warm. However, this consistent pattern has some exceptions: for example, methane fluxes did not differ from the average annual median in the hot and rainy 1995 (which little differed from 2013), although their spatial variability was as high as in 2013 and 2016. It can be assumed that the reason for the formation of the most spatially homogeneous methane fluxes among the studied elements of the microrelief is a combination of a large amount of precipitation and the air temperature below the average annual one.

**Table 7.** Medians of methane fluxes (mgC m$^{-2}$ h$^{-1}$), precipitation and air temperature in 1995–1998 and 2013–2016.

| Year | 1995 | 1996 | 1997 | 1998 | 2013 | 2014 | 2015 | 2016 | Mean |
|------|------|------|------|------|------|------|------|------|------|
| spLRV, % | 119 | 73 | 95 | 53 | 80 | 75 | 73 | 193 | 105 |
| Median CH$_4$ flux, mgC m$^{-2}$ h$^{-1}$ | 0.09 | 0.11 | 0.08 | 0.18 | 0.46 | 0.05 | 0.04 | 0.08 | 0.09 * |
| Precipitation, mm | 706 | 593 | 739 | 917 | 682 | 532 | 543 | 803 | 689 |
| Summer air temperature, °C | 14.0 | 12.8 | 12.9 | 13.0 | 14.1 | 13.5 | 13.5 | 13.7 | 13.4 |

Note: * median.

Summarizing the results obtained, we can conclude that the spatial variability will be as follows: (i) high (~220%) when high precipitation combines with high temperature; (ii) low (~50%) when high precipitation combines with low temperature; (iii) medium (~70%–120%) in other cases.

*4.4. Sources of Seasonal Variability of CH$_4$ Fluxes*

We assumed that the seasonal variability of methane fluxes can also be explained by the dynamics of precipitation and temperature (Figure 7). This assumption has been previously confirmed by some other studies in similar ecosystems [73,79–81]. The least amount of precipitation in 8 years was in 2014, as noted earlier [66], and WTL dropped catastrophically in August. The very large methane fluxes observed at the FL, DEP, and STR sites in July dropped sharply by August, following the drop in WTL. The amplitude of seasonal variability of individual methane fluxes during the 2014 season was 2 orders of magnitude there (this value of variability coincides with the data obtained earlier in [79]), while it did not exceed 500% at the EL site (Table 3). The EL sites are the most drained ones, so the catastrophic decrease in WTL did not radically change the methane flux there in August 2014. On the contrary, the methane flux changed more strongly at sites characterized by a higher WTL on average (FL, DEP, STR), likely formed in the upper soil horizons above the fallen mirror of WTL the oxidizing conditions and leading to the oxidation of a significant part of the methane which usually reached the soil–atmosphere border. The least variable during the season were the fluxes at all sites in 1996 and 2015:

the average sLRV comprised 81 and 74%, respectively. 1996 was a dry and cool year, while 2015 was dry and hot, which was the second year of this weather in a row (after 2014).

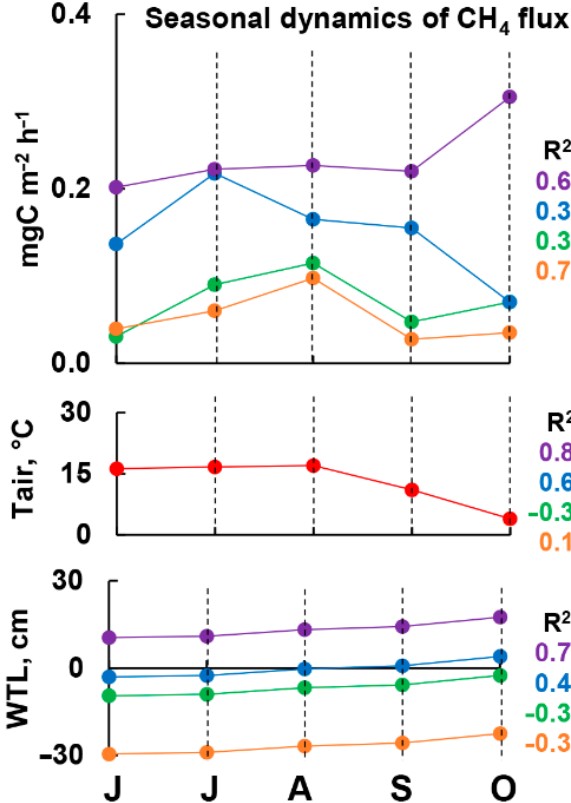

**Figure 7.** Linear regression between medians of monthly (based on all data: 1995–1998, 2013–2016) $CH_4$ fluxes (mgC m$^{-2}$ h$^{-1}$) on EL (orange), FL (green), DEP (blue) and STR (purple) sites and medians of water table levels and air temperature (multiple forward and backward stepwise—from above; separately for temperature and WTL—in the middle and at the bottom, respectively) in alder swamp "Petushikha".

The results of the regression analysis of the relationship between the dynamics of methane flux and the WTL during the season showed that the variability of the water level has the greatest direct effect on the more humid sites (DEP and STR; $R^2 = 0.7$ and 0.4 respectively), while on the drier sites (EL and PL; $R^2 = 0.3$ on both) with an increase in the WTL the methane flux is reduced. It is likely that the decrease in methane emission in dry areas with the seasonal increase in TTL is due to the more active functioning of methanotrophic communities, which become more active with the achievement of optimal soil moisture conditions [82,83]. It is possible that when the water table reached the soil surface, the methane-oxidizing layer would have appeared under conditions of oxygen deficiency and the linear regression of the methane flux from the WTL would have changed sign, as it happened at wetter sites [83,84].

Surprisingly, the dependence of the methane flux on air temperature was also greatest at the most humid sites (DEP and STR; $R^2=0.8$ and 0.6 respectively). At the site of PL, it had an inverse relationship ($R^2 = -0.3$), while at the site of EL, there was no dependence. Considering that we present the median fluxes calculated for monthly periods, the soil of the wet sites (DEP and STR) likely had time to warm up and follow the air temperature. Such dependencies have been described in numerous works [83,85,86]. The temperature dependence of the methane flux at the EL and PL sites requires further study.

Finally, multiple linear forward and backward stepwise regression showed the best regression relationship for the wettest and driest sites (DEP and EL; $R^2 = 0.6$ and 0.7

respectively). At the same time, in contrast to a separate consideration of the regression dependence only on the VTL or air temperature, the multiple regression relationship is direct at all sites.

Thus, the groundwater level, which is a consequence of the amount of precipitation, is an important factor not only in the spatial (due to position in the relief) but also in the seasonal variability of methane fluxes at least on wet sites (DEP, STR) [73,79–81], and also has some impact on sites located on a flat surface or in elevations (FL, EL). Air temperature can also have a significant direct effect on wet sites (DEP, STR) and likely the opposite on drier sites (FL).

*4.5. Sources of Interannual Variability of CH$_4$ Fluxes*

As mentioned above, the highest interannual variability was characteristic of the DEP sites, where the intraseasonal variability was also the highest among other sites. Here, as well as at the FL sites (aLRV = 93%), the least stable moistening conditions are formed over the years in comparison with other sites; for example, the EL ones that remain sufficiently drained and most of methane is oxidized in the aerated layer. At the same time, the STR sites are sufficiently moist and stably emit methane, which is reproduced from year to year, changing to a lesser extent under the influence of the amount of precipitation than the DEP and FL sites.

Unfortunately, we were unable to confirm these assumptions about hydrometeorological factors affecting the year-to-year dynamics of methane flux using a statistically significant test. We should probably study additional environmental parameters that can cause interannual variability in the methane flux; in some sources, among others, the temperature of the soil [87], vegetation cover [88], short and hard droughts [89] that were observed in the study area in 2014 and 2015 [66].

In general, the following patterns for all types of CH$_4$ flux variabilities can be formulated: i) spatial variability increases with an increase in precipitation, and to an even greater extent with a simultaneous increase in temperature and precipitation; ii) dramatic drops in WTL (by 40 cm as in August 2014 [66]) may lead to sharp decreasing of methane fluxes, affecting wet sites, which leads to a huge increase in seasonal variability; iii) finally, the lowest interannual variability is typical for sites where the WTL is either too low (EL), and most of the methane will still be oxidized regardless of the amount of precipitation, or, conversely, too high (STR), and most of the produced methane will still reach the surface soil/water.

*4.6. Optimization Methane Flux Assessment Based on Spatiotemporal Uncertainties*

The excess of sLRV over spLRV by almost 1.5 times means that the organization of flux field measurements that pursues the goals of an objective assessment of typical methane fluxes and further spatio-temporal interpretation of the data obtained should consider seasonal variability with somewhat greater emphasis than spatial variability. In other words, active precipitation in some months, or vice versa, droughts can affect the relative variability of fluxes somewhat more strongly than spatial heterogeneity and lead to underestimation or overestimation of the seasonal fluxes. At some sites, there exist "optimal" communities of methanogenic and methanotrophic organisms which are best adapted to the ecological conditions of certain elements of the microrelief; therefore, the greatest influence on methane fluxes is exerted not only by the character of their functioning, but by abrupt changes in environmental conditions which are likely to depend mainly on the meteorological situation of a particular season. On the one hand, the choice of conditions for conducting field work should fully represent the range of meteorological conditions during the season, and on the other hand, one should not neglect extreme conditions (droughts, excessive precipitation, spring snowmelt), nor overestimate them. In the best case, it is necessary to organize a permanent measurement of the amount of precipitation, WTL, and temperature, and to ensure measurements of methane fluxes in crucial (maximally different in terms of meteorological conditions) periods.

However, despite the dependence of the magnitude and variability of fluxes on environmental factors that change both seasonally and from year to year, it is necessary to form some general recommendations on the number and frequency of observations. Obviously, the variability of the means or medians will increase as the primary accumulation of temporally distributed data (such as methane fluxes). Further, the variability will stop changing as it reaches a certain threshold value since the methane flux values will begin to repeat, which itself will not increase the variability of the mean or median. That is, there is a certain threshold number of measurements performed of methane fluxes during a given period, exceeding which becomes statistically meaningless in terms of assessing the magnitude of the variability of the average or median flux. We emphasize that considering the variability of fluxes during the season does not give us an accurate idea of its seasonal dynamic; therefore, a larger number of observations is needed in order to find out not only the variability of the fluxes over the season, but also to reproduce its seasonal dynamics. A convenient way to assess the magnitude of the variability of any spatially distributed quantity is a semivariogram method:

$$\gamma(h) \;=\; \frac{1}{2N(h)} \sum_{i\,=\,1}^{N(h)} [Z(x_i) - Z(x + h)]^2 \qquad (4)$$

where $N(h)$ is the number of pairs of points separated from each other by a distance $h$, $x$ is the temporal coordinate. From a mathematical point of view, it does not matter what is meant by the distance value—therefore, we considered variability in time coordinates (days) in this case.

Figure 8 shows the methane fluxes semivariograms of the studied sites in time-variability coordinates. The threshold number of field campaigns (f.c.) averaged 7–8 in all cases (both by years and by different sites). However, the number of days on which they must be completed varies on average from 20 for EL-FL-DEP sites to 27 for STR. The optimal recommendation would be to carry out 7–8 field campaigns lasting 1–2 days every 3 days for ~3 weeks. This time and frequency may be increased for STR sites. Based on the data obtained, it can be assumed that such a frequency of measurements, performed at any time during the season, will be sufficient to characterize the variability of methane fluxes from May to October. However, as noted above, to obtain a function of the seasonal variability of methane fluxes, it will be necessary to distribute a slightly larger number of field campaigns more evenly over the season, accompanied by their accompanying measurements of key environmental drivers—in this case, WTL.

Of course, it is necessary to identify several spatially distinct sites to consider spatial variability. For example, in [65] the greatest estimated variability was of the spatial variety, although in several others (as in our case) [69,70,72] it was intraseasonal. Perhaps the optimal choice in this case would be the setup of 1–2 transects located along the gradient of natural variability of moisture conditions [90]. Based on the results of this study and other works, we assume that a natural consequence of the developed microrelief of waterlogged forests is significant heterogeneity of the aeration zone and its parcel mosaicity, which in turn results in a large spatial variation in the methane fluxes [91].

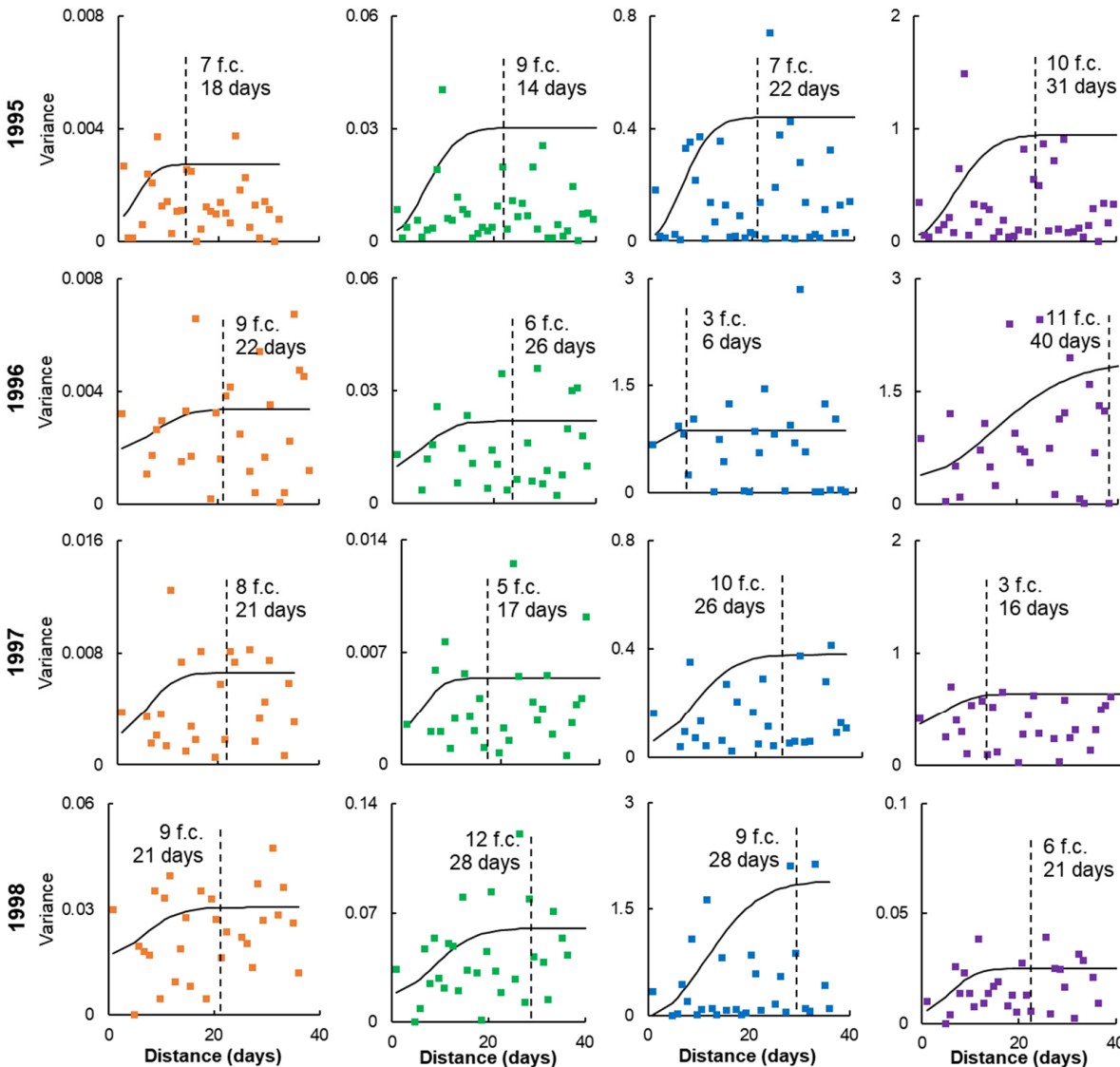

**Figure 8.** Semivariograms of methane fluxes at the sites EL (orange), FL (green), DEP (blue) and STR (purple) for the periods from 1995 to 1998, presented in time-variability coordinates. The dashed line shows the rank of the semivariogram, illustrating the number of field campaigns (f.c.) and the number of days in which they were completed, necessary to reach the threshold of variability.

Future directions of our work should focus on a more complete coverage of factors of methane fluxes variability not considered in this article, such as diurnal dynamics and spatial variability (comparison with ecosystems of other waterlogged forests). Also, despite the long-term monitoring, the issue of the seasonal pattern of dynamics of $CH_4$ fluxes remains not completely clear since it was extremely variable from year to year. The measurement of associated environmental parameters would also significantly improve understanding of the factors influencing methane flux: for example, the monitoring of water table levels, soil temperature soil moisture, etc. would allow for the parametrization of mathematical models of flux dynamics during the season, and the use of mapping methods would help to improve understanding of their spatial variability.

## 5. Conclusions

The weighted average (considering the proportion of the area of microrelief elements: DEP, FL, EL, and STR) $CH_4$ flux (1Q-median-3Q) is 0.02–0.05–0.14 mgC m$^{-2}$ h$^{-1}$. The greatest contribution to the variability of methane fluxes is made by seasonal variability (144%), somewhat smaller (105%)—by spatial, and the smallest (75%)—by interannual. We suggest that the combination of high amount of precipitation and relatively low average temperature over the summer results in lower spatial variability because of a "smoothing out" of differences in moisture conditions among microrelief elements. A large amount of precipitation and a high average temperature over the summer lead to an increase in the spatial heterogeneity of methane fluxes to a greater extent than a small amount of precipitation and a low temperature. The seasonal variability of fluxes depends to the greatest extent on the number of extreme meteorological conditions: droughts or heavy precipitation can dramatically change the magnitude of methane fluxes in short periods of time, which significantly increases the variability of fluxes. Increasing the frequency of observations during weather extremes and adequate planning of the number and location of observation sites is necessary to obtain representative estimates of the magnitude and variability of methane fluxes from sporadic sources. The next objective of our work is the quantitative assessment of the considered regularities by mathematical modeling of the dependence of methane fluxes on key environmental drivers.

**Author Contributions:** Conceptualization, S.E.V., T.V.G. and A.L.S.; methodology, S.E.V., T.V.G., A.L.S. and D.V.I.; software, D.V.I.; validation, D.V.I.; formal analysis, D.V.I., T.V.G., G.G.S. and N.A.M.; investigation, T.V.G. and A.V.G.; data curation, T.V.G. and D.V.I.; writing—original draft preparation, T.V.G., G.G.S. and D.V.I.; writing—review and editing, T.V.G., N.A.M. and D.V.I.; visualization, D.V.I.; supervision, S.E.V. and T.V.G. All authors have read and agreed to the published version of the manuscript.

**Funding:** The APC was funded by Russian Science Foundation (RSF), grant number 21-14-00076.

**Institutional Review Board Statement:** Not applicable.

**Informed Consent Statement:** Not applicable.

**Data Availability Statement:** Data is contained within the article.

**Acknowledgments:** The authors are grateful to Kovalev A.G. for conducting field studies and pre-processing the results; Zaznobin P.Yu., Denisov O.N. for assistance in arranging the infrastructure at the object of study. This research was performed according to the Development Program of the Interdisciplinary Scientific and Educational School of M.V. Lomonosov Moscow State University titled "Future Planet and Global Environmental Change".

**Conflicts of Interest:** The authors declare no conflict of interest.

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
