# Peer review of "Spatio-Temporal Variability of Methane Fluxes in Boreo-Nemoral Alder Swamp (European Russia)"

_forests, doi:10.3390/f13081178_

Round 1

Reviewer 1 Report

I find the article Spatio-temporal variability of methane fluxes in boreo-nemoral alder swamp (European Russia) interesting both from a scientific and practical point of view. The introduction is interesting and exhaustively demonstrates the need to undertake research on methane emissions from waterlogged forests. The methodological part of the article does not raise any objections. All stages of the work have been described exhaustively (I will present some minor comments below). It is similar in the case of the description of the results. They have been presented in a way that is legible to a potential reader. As previously mentioned, the article is of practical importance, especially for people who want to conduct similar research. In several places, the authors emphasized the key aspects of this type of research, e.g., the selection of the time or place of measurements. The conclusions presented at the end do not raise any objections.

Below are some small remarks that came to my mind while reading:

Lines 13-14 - I propose to give the full names of the research sites, and put abbreviations in parentheses

Lines 45-48 and 69-75 - reference to literature needed

Chapter 2.2 - I think a phytosociological records (relevés) would be better

Line 152 - please add the classification according to which the type of soil was determined

Chapter 2.4 - please specify how often samples were taken. The authors write on line 192 that the times are in the supplement, but I did not find it.

Lines 208-209 - the authors write that "The measurements 208 were carried out in the first half of the day", while in line 211, "The exposure time was 24 hours". It is a bit unclear.

Author Response

Dear Reviewer,

Thank you very much for your consideration, and we really appreciate the comments. We have tried to improve the article according to the comments made. In the file in attachment, we indicate the changes that are made in accordance with your recommendations.

Reviewer 2 Report

Please see an attachment file.

Author Response

Dear Reviewer,

Thank you very much for your attention and we greatly appreciate the comments. We have tried to improve the article in accordance with the comments made. In attachment we comment in detail on the corrections that we made to the article in accordance with the recommendations you made.
